

# No effect of low-level chronic neonicotinoid exposure on bumblebee learning and fecundity

Saija Piiroinen, Cristina Botías, Elizabeth Nicholls and Dave Goulson

School of Life Sciences, University of Sussex, Brighton, United Kingdom

## ABSTRACT

In recent years, many pollinators have declined in abundance and diversity worldwide, presenting a potential threat to agricultural productivity, biodiversity and the functioning of natural ecosystems. One of the most debated factors proposed to be contributing to pollinator declines is exposure to pesticides, particularly neonicotinoids, a widely used class of systemic insecticide. Also, newly emerging parasites and diseases, thought to be spread via contact with managed honeybees, may pose threats to other pollinators such as bumblebees. Compared to honeybees, bumblebees could be particularly vulnerable to the effects of stressors due to their smaller and more short-lived colonies. Here, we studied the effect of field-realistic, chronic clothianidin exposure and inoculation with the parasite *Nosema ceranae* on survival, fecundity, sugar water collection and learning using queenless *Bombus terrestris audax* microcolonies in the laboratory. Chronic exposure to 1 ppb clothianidin had no significant effects on the traits studied. Interestingly, pesticide exposure in combination with additional stress caused by harnessing bees for Proboscis Extension Response (PER) learning assays, led to an increase in mortality. In contrast to previous findings, the bees did not become infected by *N. ceranae* after experimental inoculation with the parasite spores, suggesting variability in host resistance or parasite virulence. However, this treatment induced a slight, short-term reduction in sugar water collection, potentially through stimulation of the immune system of the bees. Our results suggest that chronic exposure to 1 ppb clothianidin does not have adverse effects on bumblebee fecundity or learning ability.

## INTRODUCTION

In recent decades many pollinators have declined in abundance or contracted their ranges, presenting a potential threat to agricultural productivity and biodiversity (*Biesmeijer et al., 2006*; *Potts et al., 2010*). Insect pollinators are crucial for agricultural productivity providing pollination services to a wide variety of crops with an estimated global value of €153 billion per year (*Gallai et al., 2009*; *Potts et al., 2010*). Moreover, pollinators have a key role in maintaining wild plant communities with over two thirds of flowering plants being dependent on pollinators to reproduce (*Ollerton, Winfree & Tarrant, 2011*). Thus there is an urgent need to understand the underlying causes driving current pollinator declines.

Corresponding author
Saija Piiroinen,
saija.p.piiroinen@gmail.com

One of the most debated factor proposed to be contributing to pollinator declines is exposure to pesticides, particularly neonicotinoids. Widely used as seed dressings and foliar sprays for arable and horticultural crops (*Jeschke et al., 2011*), the systemic nature of neonicotinoids means that low concentrations of these pesticides may be present in the nectar and pollen of treated crops (*European Food Safety Authority, 2012*; *Sanchez-Bayo & Goka, 2014*; *Bonmatin et al., 2015*). Furthermore, residual levels of neonicotinoids can also be detected in wild flowers growing near treated crops (*Krupke et al., 2012*; *Botías et al., 2015*), meaning many flower-visiting insects may be exposed to sub-lethal doses of neonicotinoids in agricultural landscapes. However, there is a lack of consensus on whether typical levels of exposure have significant impacts on pollinators (*Goulson, 2013*). Some studies have reported no significant lethal (*Cresswell, 2011*) or sub-lethal effects of the most commonly used neonicotinoids, imidacloprid, clothianidin or thiamethoxam on honeybees (*Schmuck et al., 2001*; *Pilling et al., 2013*; *Cutler et al., 2014*) or bumblebees (*Tasei, Ripault & Rivault, 2001*; *Morandin & Winston, 2003*; *Franklin, Winston & Morandin, 2004*; *Laycock et al., 2014*) while others have indicated that exposure to sub-lethal levels of the same pesticides can cause significant harm and disruption of behaviour (*Tasei, Lerin & Ripault, 2000*; *Whitehorn et al., 2012*; *Gill, Ramos-Rodriguez & Raine, 2012*; *Laycock & Cresswell, 2013*; *Moffat et al., 2015*; *Rundlöf et al., 2015*). Acting as Nicotinic Acetylcholine Receptor (nAChRs) agonists, very low levels of neonicotinoids can disrupt neuronal functioning in bees (*Palmer et al., 2013*; *Moffat et al., 2015*), including parts of the brain essential in learning and memory (*Zars, 2000*). Impaired learning ability may be one potential mechanism underlying observed reductions in the homing ability and foraging behaviour of pesticide exposed bees (*Schneider et al., 2012*; *Henry et al., 2012*; *Gill, Ramos-Rodriguez & Raine, 2012*; *Feltham, Park & Goulson, 2014*; *Gill & Raine, 2014*).

Another potential factor driving pollinator declines is exposure to newly emerging diseases (*Meeus et al., 2011*; *Ravoet et al., 2013*). Anthropogenic movement of honeybee and bumblebee colonies has inadvertently spread bee parasites and pathogens beyond their native range (*Plischuk et al., 2009*; *Graystock, Goulson & Hughes, 2014*). For instance, European honeybees (*Apis mellifera*) are now widely infected with the microsporidian gut parasite *Nosema ceranae*, transmitted via ingestion of spores that are spread in faeces or via food exchange (*Smith, 2012*), of which the putatively original host is the Asian honeybee (*A. cerana*) (*Gómez-Moracho et al., 2015*). *N. ceranae* appears to be highly infective in European honeybees and has been shown to affect honeybee behaviour and survival (*Higes et al., 2007*; *Naug & Gibbs, 2009*). More recently, *N. ceranae* has been reported to infect also bumblebees in Europe and South-America (*Plischuk et al., 2009*; *Graystock et al., 2013*; *Fürst et al., 2014*). However, very little is known about the virulence or possible effects the parasite may pose on bumblebees (but see *Graystock et al., 2013*; *Fürst et al., 2014*).

There is a major gap in knowledge concerning the effects of field-realistic exposure to neonicotinoids on other pollinator taxa, such as bumblebees. Bumblebees play a major role in providing pollinating services to wild and crop plants, and their smaller and more short-lived colonies could make them more susceptible to the effects of stressors (*Goulson, 2010*; *Cresswell et al., 2012*; *Fauser-Misslin et al., 2014*; *Rundlöf et al., 2015*).

Here, we used bumblebee (*Bombus terrestris*) microcolonies to investigate the influence of field-realistic (1 ppb) chronic exposure to clothianidin, currently the most widely used neonicotinoid in Europe, on bumblebee survival, fecundity and feeding behaviour. Given the uncertainty whether field-realistic concentrations of neonicotinoids can have impact on bumblebee populations, and the fact that majority of studies showing detrimental effects of neonicotinoid exposure on bees have used concentrations reflecting worst-case scenarios, at the upper range of field-realistic levels, typically 10 ppb or above (e.g. *Henry et al., 2012*; *Fischer et al., 2014*; *Scholer & Krischik, 2014*), we chose to use a conservative level within the field-realistic range to better mimic the scenario the bees are most likely to experience in the field conditions. Reported values of the maximum concentrations of clothianidin residues found in nectar of treated crops vary from 1–12.2 ppb with the average values ranging from 0.3–4 ppb (*Sanchez-Bayo & Goka, 2014*; *Bonmatin et al., 2015*; *Botías et al., 2015*). Thus, 1 ppb represents a field-relevant, conservative level. We also inoculated bumblebees with *N. ceranae* to test infectivity and potential harmful effects a challenge with the parasite may pose as to aforementioned traits. Finally, both these stressors have been shown to influence brain functioning and behavioural traits in honeybees and bumblebees (*Decourtye et al., 2004*; *Gegear, Otterstatter & Thomson, 2006*; *Kralj et al., 2007*; *Aliouane et al., 2009*; *Moffat et al., 2015*; *Williamson & Wright, 2013*; *Yang et al., 2012*). Therefore, we investigated whether these factors influence learning ability and memory of individual bees using Proboscis Extension Response (PER) conditioning.

## MATERIALS AND METHODS

### Bees and pathogen screening

Fifteen nests of *B. terrestris audax* with approximately 60 workers were obtained from Biobest (Westerlo, Belgium) via Agralan Ltd (Swindon, UK). They were first screened microscopically and via PCR for common bumblebee and honeybee parasites (*Nosema bombi*, *N. ceranae*, *Crithidia bombi*, *Apicystis bombi*) by analysing a subset of workers (10% of workers) immediately upon arrival. A small sample of hind gut, mid gut and malpighian tubules was dissected out and homogenized in ddH$_2$O. DNA was extracted with 10% Chelex (Bio-Rad, Hemel Hempstead, Hertfordshire, UK). PCR protocols and parasite-specific primers followed *Graystock et al. (2013)*.

### Microcolonies and treatments

These 15 queenright nests were divided into 60 queenless microcolonies each consisting of 10 workers (4 microcolonies per queenright nest). The four microcolonies created from each queenright nest were randomly assigned to one of the four treatments: control, pesticide (clothianidin), parasite (*N. ceranae*) and exposure to both pesticide and parasite, thus we had a fully factorial design. Using microcolonies instead of whole nests enabled us to control for genetic background. Furthermore, the use of microcolonies is one of the methods recommended by EFSA for risk assessment studies (*European Food Safety Authority, 2013*). A small amount of brood and wax from the original queenright nest was given to each microcolony to stimulate nest building. Typically in a queenless

microcolony, some workers will begin laying unfertilised eggs which develop into male bees. Bees resided in circular plastic containers (diameter 11 cm, height 9 cm) with an aluminium mesh cover to allow air ventilation, and were maintained in the dark at 26 °C (±1 °C) and 55% RH (±5%).

Microcolonies assigned as pesticide or pesticide + parasite treatment were provided with an *ad-lib* supply of 60% sugar water solution contaminated with 1 ppb clothianidin (Sigma-Aldrich, Gillingham, UK) and microcolonies assigned as control or parasite treatments were given untreated food. Stock solutions were dissolved in acetone and dietary concentrations were made on the day of provisioning. Fresh sugar water solutions (contaminated with clothianidin or not) were renewed every third day, and the amount of sugar water solution collected was recorded. All microcolonies were provided with an *ad-lib* supply of untreated pollen (Biobest via Agralan Ltd, sterilised by gamma irradiation with a cobalt-60 source at dose rates between 25–45 kGy) renewed every third day.

Three days after establishment of the microcolonies, 8 bees within each microcolony individually received either a meal of 4 μl of 30% sugar water (control and pesticide treatments) or 30% sugar water containing a controlled dose of circa 130,000 *N. ceranae* spores (parasite and pesticide + parasite treatments) (viability 98.6%, viability test with 0.4% Trypan blue. Viability of the spores was confirmed by infection of honeybees). Any remaining bees in the microcolony were discarded. The rationale for only treating 8 bees was that we expected a small percentage of workers to die prior to inoculation, so each microcolony contained 2 surplus bees to ensure that an equal number per microcolony would be available to be treated. The dose administered is typical of that used in honeybee studies (e.g. *Alaux et al., 2010*; *Doublet et al., 2015*) and dosages less than 100,000 spores have been found to infect bumblebees (*Graystock et al., 2013*). Bees were starved for 2 h before treatment. To facilitate the delivery of the meal, bees were first immobilized by placing them in a cooler bag with ice blocks for approximately 10–15 min. Recovering bees ingested the inoculum when their proboscis was touched with a droplet of the spore solution at the tip of a micropipette. A solution (in ddH$_2$O) of freshly isolated *N. ceranae* spores (originating from a naturally infected honeybee hive from Spain) was obtained by homogenising abdomens of adult honeybees and purifying the homogenate by centrifugation in 95% Percoll® (Sigma-Aldrich). Identity of the parasite was confirmed by PCR. After parasite treatment, microcolonies were monitored for worker mortality and production of males every third day for 3 weeks, and then daily for 10 days. Newly emerged males were removed from the microcolonies. The number of eggs, larvae and pupae were counted at the end of the experiment. Fecundity of workers was measured as the total number of males, eggs, larvae and pupae produced during the experimental period. After the experiment, a subset of alive (n = 177) and dead bees (n = 57) were screened for *N. ceranae* infection (on average 5 bees per microcolony) as described above.

## PER assays and memory retention

Proboscis extension response (PER) assay is a standard assay of learning ability in bees, which tests their ability to learn an association between an odour (conditioned

stimulus, CS) and sugar reward (unconditioned stimulus, US) (*Bitterman et al., 1983*; *Laloi et al., 1999*). PER learning assays began 7 days after the parasite treatment (10 days after the start of chronic pesticide exposure). The day before PER conditioning started, workers were placed in a cooler bag with ice blocks for approximately 10–15 min until immobilized, before being harnessed in plastic tubes modified from 1 ml pipette tips. The head of the bee was held in place using a "yoke" made from a paper clip (see *Riveros & Gronenberg, 2009*). Harnessed bees were fed to satiation with 60% sugar solution and then starved for 15 h. In total 3 workers per microcolony were harnessed. The following day bees were first allowed 20 min to acclimate to the conditions of the room in which the PER assays were conducted (mean temp 23 °C). Then one bee at a time was placed at 5 cm distance from a continuous air flow (circa 2 L min$^{-1}$, aquarium pump, Hidom, Shenzhen, China) delivered by a silicon tube (diameter 4 mm) with a 1 ml pipette tip at the end. A floral odour (CS) linalool (Sigma-Aldrich) was then delivered by switching the air flow pass through a 20 ml syringe containing a $2 \times 20$ mm filter paper with 5 μl 2 M of the odorant in mineral oil. An extractor fan was placed behind the bee to allow extraction of any residual odour. One PER assay was composed of 10 CS-US trials with each trial conducted as follows: 15 s air flow, 3 s CS (air flow with odour only), 3 s both CS and US (unconditioned stimulus: touching the antennae and proboscis with a toothpick dipped in 60% sugar solution (not spiked with pesticide) and allowing the bee to lick), 1 s US only, 8 s air flow. The Inter-Trial Interval (ITI) was 10 min. Extension of the proboscis during each CS and US was recorded. After 10 CS-US trials, a final level of learning acquisition was assessed by recording whether a bee extended its proboscis when presented with only the conditioned stimulus without the sugar reward (US).

To test whether bees remembered the learned association, a memory retention test was conducted 2 h after each PER assay, where again only the conditioned stimulus was presented, in the absence of the US. Ten minutes before and after each PER assay and after each memory retention test, bees were tested for motivation to respond to sugar stimulus by touching the antennae with a toothpick covered with 60% sugar solution and observing whether extension of proboscis occurred. Bees that showed a negative response to the sugar solution were excluded from the analyses. Also, bees that showed 4 or more sequential negative responses to US during PER assays were considered unmotivated and were excluded from analyses. After testing memory retention, bees were released from the harness, returned to their microcolony and their survival was observed.

## Statistics

We used generalized (GLMM) and linear (LMM) mixed effect models in IBM SPSS v 21 (IBM SPSS Inc., USA) to analyse the effect of pesticide exposure and parasite treatment on measures of learning, fecundity and sugar water collection. For models with repeated measures structure (learning performance, sugar water collection), bee or microcolony identity was added as subject and trial number (trials 2–10) or the time point of sugar water measurement (6, 9, 12, 15, 18, 21 and 24 days) as a repeated variable. In the analysis of learning performance, the first trial of the 10 CS-US trials was omitted to increase

model fit (because it contained only zeroes). Binomial error structure with a logit link function was used in models analysing learning performance, final acquisition of learning (11th trial) and memory retention. A negative binomial error distribution with a log link was used to account for overdispersion in the models when the response variable was number of positive responses to CS and fecundity. In the analysis of sugar water collection, assumptions of homogeneity and normality of residuals were checked by inspecting residual plots (residuals against predicted values) and qq-plots. Bumblebee queenright nest (the original nest from which workers were divided into microcolonies) was considered as a random variable in the models. We first fitted the full model after which interaction terms were omitted if they did not increase the model fit based on Akaike Information Criterion (AIC). AIC was also used in selecting the repeated covariance type in models with repeated measures structure. Significant interactions were post hoc tested with simple effects tests.

Worker survival (dead, alive at the end of the experiment) was analysed with a Generalized Linear Mixed Effects Model (GLMM) with Laplace estimation and a logit link function in R (version 3.1.3) (*R Development Core Team, 2015*) using the glmer function from the package lme4 (*Bates et al., 2014*). Pesticide exposure and parasite treatment as well as harnessing (harnessed for PER or not) were entered as fixed variables. Bumblebee queenright nest and microcolony were included as random effects. We first fitted the full model, interaction terms were then removed if they did not significantly increase the model fit based on maximum likelihood ratio test. Significant interactions were further examined in post hoc tests performed within each factor.

## RESULTS

### Nosema inoculations

PCR screenings of a subset of alive and dead bees (38.3%, 36.6%, 55.8%, and 64.1% of the bees in the control, pesticide, parasite and pesticide + parasite groups, respectively) showed that only 1 bee belonging to the parasite treatment group was infected with *N. ceranae*. No spores were detected under the microscope. Due to potential stressful effects ingested parasite spores may exert on bees, even though the bees were able to resist the infection (see discussion), this parasite treatment is treated as a separate group in the results reported below.

### PER and memory retention

Fifty four percent (97 out of 180) of harnessed bees successfully completed PER training. 19.4% of bees died during the starvation period and the remainder 26.7% were not sufficiently responsive to the US stimulus. Learning performance, measured as the proportion of positive proboscis extension responses to the odorant, Conditioned Stimulus (CS), at each trial, increased during the 10 CS-US trials reaching 59.7% overall at the 10th conditioning trial (Tables 1 and 2; Fig. 1). Pesticide exposure or parasite treatment (note, bees did not become infected) did not affect learning performance (Table 1). The motivation of bees to respond to the US stimulus remained high throughout the 10 CS-US trials increasing overall from 82.5%–96.9% (GLMM,

**Table 1 Results of repeated generalized linear mixed effect model (GLMM) on the learning performance of *B. terrestris* workers chronically exposed to pesticide clothianidin and ingested parasite *N. ceranae* spores.**

| Effect | F | d.f. (n, d) | P |
|---|---|---|---|
| Corrected model | 59.41 | 3,869 | <0.001 |
| Pesticide | 0.46 | 1,869 | 0.50 |
| Parasite | 0.55 | 1,869 | 0.46 |
| Trial number | 178.14 | 1,869 | <0.001 |

Note:
Restricted maximum-likelihood procedure was used in the estimation. Random factor nest, estimate ± residual: 0.28 ± 0.21, z = 1.30, P = 0.19. Akaike information criterion (AIC) = 3666.30, repeated covariance type: compound symmetry. Interaction between pesticide and parasite was non-significant (F(1,867) = 0.20, P = 0.66) and was removed during model selection.

**Table 2 Percentage of *B. terrestris* workers showing positive proboscis extension response (PER) and total sample sizes at the 10th CS-US trial, 11th CS trial (final level of learning acquisition) and after 2 h in the memory retention test for different treatment groups.** Pesticide: bees chronically exposed to pesticide clothianidin. Parasite: bees ingested parasite *N. ceranae* spores. Both: bees chronically exposed to pesticide clohianidin and ingested parasite *N. ceranae* spores.

| Treatment group | 10th Trial % PER (n) | Final level of learning acquisition % PER (n) | Memory retention test % PER (n) |
|---|---|---|---|
| Control | 54.5 (22) | 68.2 (22) | 23.8 (21) |
| Pesticide | 69.6 (23) | 69.6 (23) | 36.8 (19) |
| Parasite | 57.7 (26) | 69.2 (26) | 43.5 (23) |
| Both | 57.7 (26) | 65.4 (26) | 31.8 (22) |

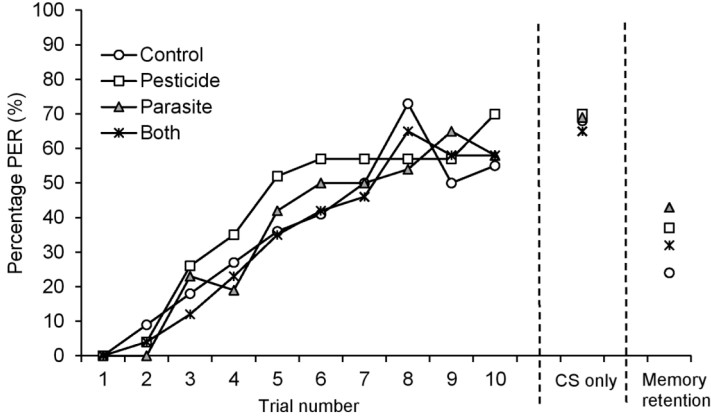

**Figure 1 Percentage (%) of *B. terrestris* bumblebee workers showing proboscis extension response (PER) to odour (CS, conditioned stimulus) stimulation across 10 CS-US (US: unconditioned stimulus) trials for bees exposed to pesticide clothianidin and ingested parasite *N. ceranae* spores.** CS only: percentage of PER at the 11th CS trial. Memory retention: percentage of PER 2 h after learning acquisition.

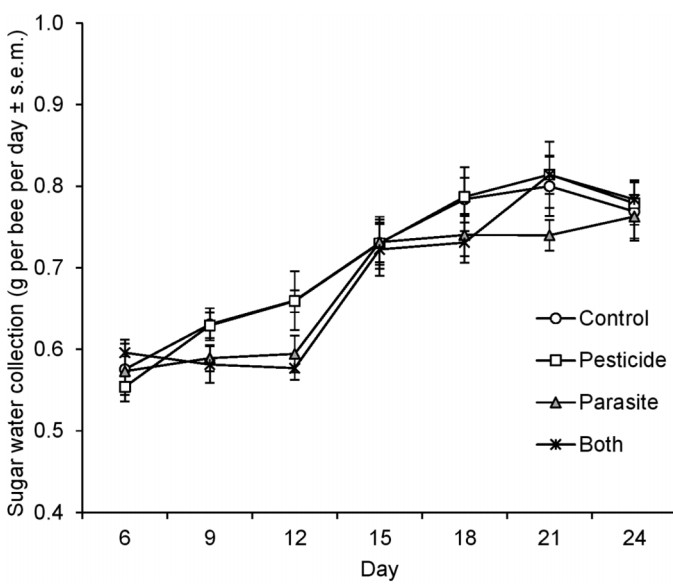

**Figure 2 Sugar water collection of *B. terrestris* bumblebee microcolonies exposed to pesticide clothianidin and ingested parasite *N. ceranae* spores.**

F = 35.32, d.f. = 1, 966, P < 0.001). There was no difference among treatment groups in the motivation to respond to the US (GLMM, Pesticide: F = 3.012, d.f. = 1, 966, P = 0.09; Parasite: F = 1.28, d.f. = 1, 966, P = 0.26).

Overall, 68% of bees showed a positive proboscis extension response in the final test, where bees were presented with the CS-only (trial 11, final acquisition level, Table 2; Fig. 1). There was no difference among treatment groups in the final level of acquisition (GLMM, Pesticide: F = 0.002, d.f. = 1, 94, P = 0.97; Parasite: F = 0.01, d.f. = 1, 94, P = 0.91) or in the total number of positive PERs to CS (GLMM, Pesticide: F = 0.11, d.f. = 1, 94, P = 0.74; Parasite: F = 0.28, d.f. = 1, 94, P = 0.60).

Overall, 34% (n = 85) of the tested bees remembered the association between the odour and sugar reward when tested 2 h after PER conditioning (Table 2; Fig. 1). There was no difference in memory retention among treatment groups (GLMM, Pesticide: F = 0.002, d.f. = 1, 82, P = 0.96; Parasite: F = 0.68, d.f. = 1, 82, P = 0.41).

## Sugar water collection

There was a significant interaction between parasite treatment and time point on sugar water collection by bees (LMM, F = 4.75, d.f. = 6, 58, P = 0.001, repeated covariance type: unstructured) indicating that the influence of ingestion of *N. ceranae* spores on sugar water collection was time-dependent (Fig. 2). While sugar water collection increased with time, it did not differ between parasite treated and control microcolonies at the beginning of the experiment (at 6 days: P = 0.21). However, parasite treated microcolonies collected less sugar water than control ones at measurement days 9 and 12 (both, P = 0.003) after which no differences were observed (all time points, P > 0.32, Fig. 2). Pesticide exposure did not affect sugar water collection (F = 0.23, d.f. = 1, 41.4, P = 0.64).

**Table 3 Results of a generalized linear mixed effect model (GLMM) on survival of *B. terrestris* workers exposed to pesticide clothianidin and ingested parasite *N. ceranae* spores, and harnessed for proboscis extension response (PER) assay.**

| Effect | Estimate | s.e.m | z | P |
|---|---|---|---|---|
| Pesticide | 0.76 | 0.48 | 1.61 | 0.11 |
| Parasite | −0.08 | 0.37 | −0.23 | 0.82 |
| Harnessing | −2.24 | 0.37 | −6.06 | <0.001 |
| Pesticide × Harnessing | −1.46 | 0.54 | −2.74 | 0.006 |

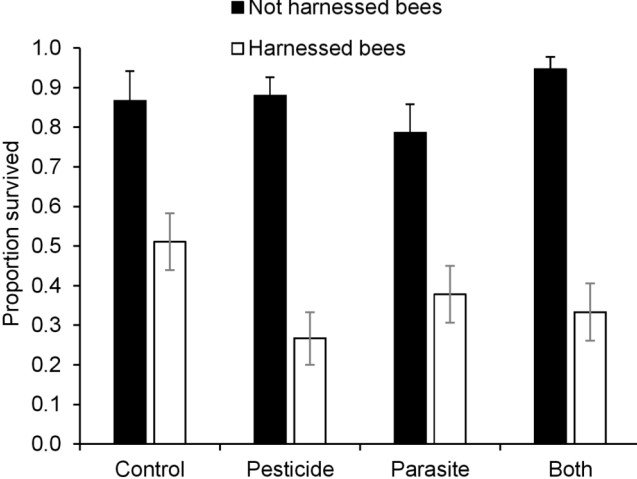

**Figure 3 Survival (mean ± s.e.m) of *B. terrestris* bumblebee workers harnessed and not-harnessed for proboscis extension response (PER) analysis, and exposed to pesticide clothianidin and ingested parasite *N. ceranae* spores.**

## Fecundity

Neither pesticide exposure nor parasite treatment affected fecundity (total number of males, eggs, larvae and pupae produced by workers in microcolonies) (GLMM, Pesticide: $F = 1.24$, d.f. = 1, 57, $P = 0.27$: Parasite: $F = 0.55$, d.f. = 1, 57, $P = 0.46$). Fecundity was on average 99.27 ± 13.44 (1 s.e.m.), 111.67 ± 14.90, 89.07 ± 10.80 and 101.47 ± 12.44 for control, pesticide, parasite and pesticide + parasite groups, respectively.

## Survival

There was a significant interaction between pesticide exposure and harnessing (Table 3) indicating that the influence of pesticide exposure on survival depended on whether bees were harnessed or not for PER assays (Fig. 3). Post hoc analysis revealed that pesticide exposure decreased survival of bees that were harnessed but not in bees that were not harnessed (Harnessed: $z = −2.00$, $P = 0.04$: Not harnessed; $z = 1.33$, $P = 0.18$). In general, bees that were not harnessed for PER assay had much higher survival than those harnessed for PER. Parasite treatment did not affect survival of bees ($z = −0.23$, $P = 0.82$).

## DISCUSSION

Given the importance of pollinators on ecosystem services, there is an urgent need to understand the causes of pollinator declines (*González-Varo et al., 2013*; *Goulson et al., 2015*). We studied the effect of chronic exposure to field-realistic sub-lethal levels of clothianidin on bumblebee learning, fecundity and survival. Exposure to 1 ppb clothianidin in sugar water had no significant effects on the traits studied other than an increase in mortality when combined with the stress of harnessing for learning assays.

Our results are in line with some previous studies that have observed no adverse effects of sub-lethal, field-realistic exposure of neonicotinoids on bumblebee mortality or reproduction. Exposure of *B. impatiens* colonies to 6 or 36 ppb of clothianidin (8 weeks in pollen/sugar water mixture) did not cause any significant reduction in the production of brood, workers, males or queens (*Franklin, Winston & Morandin, 2004*). Also, sub-lethal exposure to another neonicotinoid compound, thiamethoxam, did not cause adverse effects on the reproduction or mortality at exposure levels below 10 ppb (28 days in pollen and sugar water: *Elston, Thompson & Walters, 2013*, 17 days in sugar water: *Laycock et al., 2014*) in contrast to imidacloprid (12 weeks in pollen and sugar water: *Tasei, Lerin & Ripault, 2000*, 13–14 days in sugar water: *Laycock et al., 2012*; *Laycock & Cresswell, 2013*), to which bumblebees may be more sensitive (*Laycock et al., 2014*). However, there is contrasting evidence as to whether sub-lethal exposure to thiamethoxam or to its major metabolite, clothianidin, has harmful effects on bumblebees. *Fauser-Misslin et al. (2014)* found that exposure for 9 weeks to 1.5 ppb clothianidin and 4 ppb thiamethoxam reduced worker production and longevity of bumblebee colonies in the laboratory. Furthermore, a recent field experiment by *Rundlöf et al. (2015)* revealed that exposure to clothianidin in seed coated oilseed rape was associated with lower wild bumblebee densities and negatively affected colony growth and reproduction in commercially reared bumblebee nests (monitored for 40 days). The mean concentration of clothianidin in bee-collected nectar was 5.4 ppb (range 1.4–14 ppb) (*Rundlöf et al., 2015*), which is higher than that used in our study (1 ppb) and higher than residue levels typically found in nectar of treated crops (*Godfray et al., 2014*; *Sanchez-Bayo & Goka, 2014*; *Bonmatin et al., 2015*). Overall, it appears that exposure to 1 ppb clothianidin used in our study is not harmful to bumblebees in a laboratory setting, whereas slightly higher exposure levels combined with other environmental stressors can result in impaired fitness effects (*Fauser-Misslin et al., 2014*; *Rundlöf et al., 2015*). It is also worth noting that we only spiked sugar water and used only one neonicotinoid whereas *Fauser-Misslin et al. (2014)* also spiked pollen and exposed bees to two different neonicotinoids.

Even though reproduction was not affected, sub-lethal effects of pesticide exposure may be manifested in other traits such as behaviour, where subtle changes in behaviour or learning may result in significant impacts at the colony level (*Raine & Chittka, 2008*). However, our PER assays did not reveal any significant effects on learning or memory at the dose used. Recently, a laboratory study exposing bumblebees to as low as 2.5 ppb thiamethoxam for 3.5 weeks was found to impair olfactory learning (*Stanley, Smith & Raine, 2015*). In honeybees, both negative and positive effects of pesticides on olfactory

learning ability have been reported (*Decourtye et al., 2004*; *Williamson & Wright, 2013*; *Williamson, Baker & Wright, 2013*). For instance, four days chronic exposure to sub-lethal concentrations of imidacloprid (10 nmol l$^{-1}$ ~ 2.3 ppb) reduced learning acquisition in *A. mellifera* (*Williamson & Wright, 2013*) and a single acute dose of 0.1 ng imidacloprid was sufficient to impair learning acquisition in the Asian honeybee *A. cerana* (*Tan et al., 2015*). Interestingly, in our study, pesticide exposed bumblebees had slightly, but not significantly, faster acquisition rate compared to the other treatment groups, suggesting possible hormetic effects (*Cutler & Rix, 2015*). It would be useful to follow up our study by studying learning of bumblebees when exposed to a range of concentrations of pesticide, although it should be noted that PER assays are very labour intensive to perform.

Interestingly, pesticide exposure in combination with additional stress caused by harnessing bees for 15 h and conducting learning assays increased mortality. Overall, PER tested bees had higher mortality compared to those not harnessed and tested in the PER assay. It is not clear whether cold stress, harnessing, starvation prior to the assay, some facet of the PER test itself, or possibly a combination of these factors results in a higher risk of mortality. It is known that cold exposure can be stressful for the bees (*Poissonnier, Jackson & Tanner, 2015*). However, only 19.4% of the bees died during the 15 h starvation period (after cold narcosis), the majority of bees (37.2%) dying within the three days following PER testing. This indicates that cold exposure and/or starvation stress did not cause high immediate mortality, but nonetheless may have inflicted measurable harm leading to delayed mortality, particularly in bees also exposed to pesticides. The possible interactive effects of pesticide with additional stress observed here would demand further investigation.

Rather unexpectedly, the bumblebees in our study did not become infected by *N. ceranae*. There are only few studies that have experimentally investigated virulence of *N. ceranae* in bumblebees with infection levels shown to vary substantially, from about 30% (*Fürst et al., 2014*) to over 50% (*Graystock et al., 2013*), with survival ranging from 100% to 35%, respectively. It could be that the *N. ceranae* strains used in our study were either not infective towards bumblebees, or that the commercial bumblebee strains used were resistant to this gut parasite. Both the host and the parasite play an essential role in determining the outcome of an infection (*Frank & Schmid-Hempel, 2008*) in which the nature of fitness trade-offs between them may determine the level of virulence. At the same time, *Nosema* strains vary in infectivity and virulence (*Genersch, 2010*) and several studies have reported different degrees of tolerance and resistance to *N. ceranae* associated with the genetic background of the honeybee host (*Dussaubat et al., 2013*; *Fontbonne et al., 2013*; *Huang et al., 2013*; *Huang et al., 2014*). It should also be noted that the *Nosema* strain we used in our study was able to infect honeybees inoculated with the same dose as used in the experiment (results not shown), indicating that the very low infectivity in bumblebees is not because the spores were inviable.

Even though bees were not infected by *N. ceranae*, the parasite inoculation had slight, short-term effect on sugar water collection. Reasons for reduced collection of sugar water 6–9 days after *Nosema* treatment (9–12 days after starting pesticide feeding) are not clear. In honeybees, *Nosema* challenge has been linked to increased energetic stress

leading to increased feeding rates (*Naug & Gibbs, 2009*; *Mayack & Naug, 2009*; *Martín-Hernández et al., 2011*). However, as the inoculation of the bumblebees with viable *N. ceranae* spores did not lead to infection in our experiment, the same outcome cannot be expected. Interestingly, previous studies have observed stimulation of the bumblebee immune system just a few hours after ingesting food containing pathogenic microorganisms (*Riddell et al., 2011*; *Brunner, Schmid-Hempel & Barribeau, 2013*), and even when fed non-pathogenic elicitors such as lipopolysaccharides (*Moret & Schmid-Hempel, 2000*). Therefore, we postulate that the short-term reduction in the collection of sugar water in microcolonies treated with *Nosema* may be due to some behavioural or physiological alterations related to the stimulation of the immune response in the host after contact with the parasite (*Alghamdi et al., 2008*), though further research would be needed to confirm this. Overall, our results on *Nosema* emphasizes the need for further research on infectivity and virulence of *N. ceranae* in bumblebees in order to assess whether *N. ceranae* presents a serious threat to bumblebee health.

## ACKNOWLEDGEMENTS

We thank Dezene Huber and three anonymous reviewers for their constructive and helpful comments, and are grateful to Mariano Higes, Raquel Martín-Hernández, Cristina Rodríguez-García and Teresa Corrales from CIAPA Research Institute for providing *Nosema* spores.

### Funding

This work was financially supported by Jenny and Antti Wihuri Foundation Post-Doc Pool. The funders had no role in study design, data collection and analysis, decision to publish, or preparation of the manuscript.

### Competing Interests

The authors declare that they have no competing interests.

### Author Contributions

- Saija Piiroinen conceived and designed the experiments, performed the experiments, analyzed the data, wrote the paper, prepared figures and/or tables, reviewed drafts of the paper.
- Cristina Botías wrote the paper, reviewed drafts of the paper.
- Elizabeth Nicholls wrote the paper, reviewed drafts of the paper.
- Dave Goulson conceived and designed the experiments, wrote the paper, reviewed drafts of the paper.

### Data Deposition

The raw data can be found in the Supplemental Information.
## Supplemental Information

Supplemental information for this article can be found online at http://dx.doi.org/10.7717/peerj.1808#supplemental-information.

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
