# Peer review of "No effect of low-level chronic neonicotinoid exposure on bumblebee learning and fecundity"

_PeerJ, doi:10.7717/peerj.1808_

## Round 0.1 · original submission · Major Revisions

Thank you to both reviewers for their extensive and comprehensive reviews of this MS. Both have given a number of items that the authors should pay careful attention to.

I agree with Reviewer #2 that the failure of the Nosema infection to take hold (except for in one bee) needs to be addressed further by the authors. The lack of infection does shift the design to some extent in that one might argue that there were really effectively two treatments, not four. On the other hand, if there was some level of resistance to the infection in the population of bees used, there still were four treatments as there would have undoubtedly have been some energetically and otherwise costly immunological response.

In any case, the authors should reevaluate how they deal with the lack of infection. Several suggestions are:

1. Move the failure to obtain infection to the top of the results section so that readers know what they are dealing with as they contemplate the rest of the results.

2. I think that #1 would help to tone down the Nosema-based result a bit in the rest of the results, but the authors should consider further measures.

3. The fact that no infection was obtained is, however, interesting in its own right and might indicate (as intoned in the MS) a resistant bumblebee strain and/or an infective agent strain that was not capable of infecting bumblebees. I assume the authors have further information on the bumblebee and Nosema strains, and so should include that information in the MS as it would be potentially useful for further work in this regard.

4. Is it possible that there were issues with the inoculation method? The authors should be sure to be very details about the methodology here. I believe that they generally were, but perhaps there is more detail that could be added? If followup studies are to be done, detail is vital. And notes of "failure" in the literature can be helpful for others designing studies to replicate or expand on other work.

It is interesting to note that the stress of handling for the PER work interacted with the neonic treatment to increase mortality. While other stressors might have been considered in other treatments, the collection of that data is important as well for further work in this type of system. I.e., researchers should keep potential interactions of this sort in mind and not allow such interactions to confound results on other stressors.

As noted by review and in the MS itself, the neonic dose was at the low end. This is not a problem as I see it, as it is noted in the MS (and even in the title). However the authors should go into more detail (including numbers, context, and references) in terms of why such a low level is warranted in terms of in-the-field reality.

In general the authors are reporting negative results. This is fine, and in fact, ought to be encouraged more in the literature. Both reviewers find the experimental methods, as described, adequate. I do as well. The conclusions, for the most part, do not overreach the data. As such, this paper has merit, but the authors need to provide a robust rebuttal and, depending on the outcome at that stage, I may seek a second round of review.

·

Basic reporting

Appropriate.

Experimental design

Appropriate

Validity of the findings

No concerns

Additional comments

Overall well done. An interesting and valuable study in that: (1) the results contrast with those of other laboratory studies showing deleterious effects on Bombus learning at slightly high concentrations of neonicotinoid insecticide; and (2) Nosema ceranae may not be infective of Bombus.

Specific comments:
1. L14. Change to “potential threat”. There is little current evidence that pollinator declines are a threat to agriculture. In fact, just the opposite seems to be true as production of pollinator-dependent crops in increasing, not decreasing.
2. L18. Remove “serious”.
3. L27-28. Be precise - what was the effect on sugar water collection?
4. L34. Change to “potential threat”
5. L44. Change to “….low concentrations of these pesticides may be present in the nectar…” Often these compounds are not detected at all (below the LOQ/LOD), and this varies with crop and agronomic conditions.
6. L47. Botias et al 2015 found wildflowers outside of the OSR neonicotinoid-treated field with pollen residues >20 times greater than the treated area and >100 levels found in the soil of the field margin – how is this possible?!
7. L53-54. Also consider these references that show no effects: Pest Manag Sci 56:784–788; J Econ Entomol 94:623–627; Environ Entomol 32:555–563; Ecotoxicology (2014) 23:1755–1763
8. L55. Tasei et al (2000) taken out of context. Quote from the conclusion of the abstract of Tasei et al (2000): “It was concluded that the survival rate and reproductive capacity of B terrestris was not likely to be affected by prolonged ingestion of nectar produced by sunflower after seed-dressing treatment with imidacloprid”
9. L111. Clarify that these are the same colonies described on L101
10. L123. Change to “Microcolonies assigned as pesticide or pesticide+parasite treatment… “
11. L126. Stock solutions containing Nosema were dissolved in acetone as well? Would this effect infectivity?
12. L151. Clarify how many alive vs dead bees were collected and how many approximately were collected from each microcolony.
13. L167. Specify the floral odour and its source
14. L185. This is clearly a very stressful event for bees. Given that bees were returned to microcolonies, I wonder if the stress experienced by the PER negated any potential stressful effects of the pesticide or parasite, including the controls.
15. L185. Fig 3 indicates that survival of these bees was tracked. Add this detail.
16. L201, 211. “Bumblebee queenright nest” refers to the commercial colonies? If so, good, but clarify.
17. L205. “tests” should be plural.
18. L215. Great description of the data analyses.
19. L224, 232, 238. Was not the effect of pesticide and parasite compared to the control, not each other? Instead of saying “there was no difference between the…”, should this not be “There was no effect of …..”?
20. L263. Change to “decreased survival of bees…”
21. L296. Worth noting that the levels found by Rundlof et al were noticeably higher than typical levels detected in pollen and nectar. E.g. Godfray et al (2014) state: “Estimates of the concentration of neonicotinoids in the pollen and nectar of seed-treated crops vary considerably with AVERAGE MAXIMUM levels (from 20 published studies) of 1.9 (nectar) and 6.1 (pollen) ppb”. i.e. the mean levels in pollen and nectar from Rundlof are >2-fold and >5-fold, respectively the mean MAXIMUM of 20 other studies
22. L301. More importantly, Fauser-Misslin et al., 2014 exposed bees in the lab continuously for 9 weeks – very unlikely this would occur in the field.
23. L307-308. Clarify that Stanley et al (2015) was a lab study that continuously force-fed bees pesticide treatment for 3.5 weeks.
24. L333. I wonder would the same results be seen if agents more specific to Bombus, such as Nosema bombi and Crithidia, were used instead. Would be worth mentioning this.
25. L336. What was the power of the experiment? I am not suggesting the authors do a power analysis (although this might be valuable) but if the experiment were repeated would they expect to see the same effect at days 6-9?
26. Table and fig captions. Some of these need more detail. E.g. Table 1 title is fine as it gives a clear indication of the test organism, specific treatments, and analytical methods, but most of these details are missing from the titles of Tables 2 and 3.
27. Fig 1-3. Captions should indicate specific type of pesticide and pathogen; no need to indicate “g per bee per day ± s.e.m” in Fig 2 caption since this is indicated on the y axis; Fig 1 – define PER, US, CS; Fig 3 – define PER

Reviewer 2 ·

Basic reporting

The manuscript is well written and cites the appropriate literature. However, the use of Nosema infection should be removed as it did not work.

Experimental design

The experimental design was good, but unfortunately the failure to get Nosema infection destroyed the design, leaving the authors to make the best of what data they had.
Given the range of clothianidin found in the field (1-12.2 ppb), it is a shame that the authors limited their study to the lower end of this range, especially when they observed no effect. Repeating this study with the inclusion of 12.2 ppb would more usefully reflect any environmental risk from normal use of clothianidin.

In the same vein, the author should include the use of thiamethoxam as this has been demonstrated to have a chronic effect on learning and memory (Stanley et al 2015) and expecially as it is inactive until metabolized to clothianidin. Therefore, it is an essential positive control to determine if the findings are valid or a consequence of methodological differences.

Validity of the findings

Unfortunately, the group has been unsuccessful in infecting bumble bees with Nosema. As a consequence, this aspect of the study needs to be removed from the manuscript (or repeated to get successful infection). Despite it failing, it still exists as a large component in this manuscript, especially the final discussion where it is irrelevant.

In the PER experiments, it is not clear whether clothianidin exposure is acute or chronic (24 days). In Stanley et al (2015), no acute effects were observed for thiamethoxam, therefore chronic exposure should be examined here too, before statement of no effect can be made with confidence.

With respect to fecundity and survival, it is well established that neonicotinoids are not lethal to bees at field-relevant levels and I am not sure that too much emphasis should be placed on worker bee egg laying. Queen laying would be more relevant but obviously more expensive to perform.

The interaction between clothianidin and harnessing on mortality is interesting but completely artefactual (it doesn’t happen in real life). Being queenless may be stressful enough to affect the baseline in these studies. Clearly, the point of the study was to use Nosema infection as the stress inducer and this would be very interesting indeed. As the harnessing didn’t impact learning, it is hard to reconcile it with a stress-inducer. Therefore, the data is tenuous. It would be much better if the original approach could be delivered, or replace with another stressor.

Additional comments

In Stanley et al (2015) they performed PER on bumble bees exposed chronically to thiamethoxam for 21 days and reported no mortality but a significant impact on learning and memory. Given that thiamethoxam is inactive until metabolized to clothiandin (here clothianidin is fed for 24 days), the results should be similar. To demonstrate that clothianidin does not impact bee survival and fecundity, it will be necessary to repeat the study to include 12.2 ppb clothianidin and also thiamethoxam (2.4 ppb). If differences remain, then the central dogma is incorrect. If they are the same (but different from Stanley et al 2015) then methodological explanations should be sought.

As this manuscript stands, it feels like a re-working of a failed experiment to re-focus on what did work. Unfortunately, this does not work with the existing dataset. I believe that a good study could be generated by a little further work. To make this study valid there are several improvements required: 1. To get Nosema infection or to remove it from this manuscript. 2. Clothianidin needs to be examined at higher field-realistic levels to determine if it really has no impact on bumble bee learning, memory and mortality. 3. Thiamethoxam needs to be included (2.4 ppb, 21 days) to demonstrate that its effects (Stanley et al 2015) do occur under the methodology used here (a positive control is essential). 4. In the absence of Nosema infection, an alternative stress (eg. cold, hunger, lack of pollen) needs to be validated and applied to investigate interactions with neonicotinoids.

---

## Round 0.2 · Minor Revisions

Reviewer #3 wrote the following:

"I agree with the authors' decision to treat the 'infection group' in the design because they were treated differently from the other groups."

...and I think that this is a key point here.

Reviewer #1 wrote:

"The fact that a N ceranae strain shown to be infectious to Apis was not infectious to Bombus in the current experiments, is interesting and worth reporting since others have recently shown that N ceranae can be infectious to Bombus."

...again, an important point.

Following the very minor revision (exposure route + duration as found in the literature) suggested by Reviewer #3, this paper should be ready for publication in PeerJ.

Thank you for your contribution.

·

Basic reporting

No concerns

Experimental design

Adequate

Validity of the findings

No concerns

Additional comments

I have reviewed the revised manuscript and responses put forth by the authors. I am satisfied with the responses to my questions and concerns‎, and contend that the paper is acceptable for publication. ‎I also agree with the authors and the Editor that the 'negative' results should not preclude publication of the paper or certain experiments. Indeed, this is precisely why this particular paper has value. Too often researchers are bent on reporting 'positive ' results, and will design experiments in order to do so, at the cost of poor a priori rationale for treatment scenarios. This is very true in many toxicology studies with pesticides and bees where exposure concentrations and durations are often unrealistically high. In the current experiments, the authors have avoided this and have used a 'typical' exposure concentration (as confirmed in many previous studies, although in MANY instances exposure concentrations will be even lower still), not atypical outlier/worst case exposure concentrations and durations that bees seldom encounter in the field. The only potential issue with the negative results from a single exposure concentration is that one could argue the method has not been validated with a positive control.

The fact that a N ceranae strain shown to be infectious to Apis was not infectious to Bombus in the current experiments, is interesting and worth reporting since others have recently shown that N ceranae can be infectious to Bombus. That is, it is important that readers and scientists be reminded that previous reports of Bombus being susceptible to N ceranae infection may be the exception, not the rule. The authors have rightly pointed out in their concluding sentence that more work is needed in this area.

Reviewer 2 ·

Basic reporting

The paper reads much better now.

Experimental design

There are 2 distinct pieces of work here.
1. The effects (or lack of) of clothianidin and its interaction with stress.
2. Failure to infect B. terrestris with N.ceranae .

The first study is very interesting as it contrasts with the negative effects reported for imidacloprid and thiamethoxam. However, these are not compared. I think this section should be expanded to include the other 2 neonicotinoids and also to employ a more natural stress, such as lack of pollen or low temperature. This would make a very interesting and important study and should be submitted as an independent piece of work.
The second aspect of the study finds that B. terrestris could not be infected with N. ceranae. Again this is very interesting but needs more evidence to be conclusive. For example could this still re-infect honeybees (ie a positive control) and is there a possible strain difference (bee or Nosema) that might account for this failure. Alternatively, there is evidence that N.ceranae is an opportunistic infection and bees may need to be stressed before infection occurs (they may just not have a productive, sporulating, infection). This would also be a very interesting second paper.

However, together the 2 half stories do not make a single one.

Validity of the findings

More evidence (as reported under 'experimental design') is required to be sure that:
1. clothianidin, unlike imidacloprid and thiamethoxam, has no effect.
2. A realistic stress renders bees vulnerable to pesticide toxicity.
3. N. ceranae cannot infect bumblebees.

Additional comments

I think that you have 2 very interesting, but partial, stories that should be worked up into 2 independent manuscripts and I would be very happy (and supportive) to see this in review. I think that much of the potential interest will be lost if these stories are mixed into a single weak study. Only if successful N.ceranae infection is achieved in B. terrestris should the 2 stories be merged (I'm sure there would be an interaction).

Reviewer 3 ·

Basic reporting

The manuscript did an excellent job of describing the hypotheses tested, their justification, how the data was collected and analyzed, the main findings of the paper, and the conclusions.

Experimental design

The experimental design was sound. i agree with the authors' decision to treat the 'infection group' in the design because they were treated differently from the other groups.

Validity of the findings

I think the authors were careful in interpreting their data. and felt that the conclusions did not outstrip their findings.

Additional comments

This is minor but important: in the discussion (and also to some extent in the introduction) the authors compare different treatment studies by only referring to the treatment dose: e.g. X et al fed bees 10 ppb of a pesticide and found an effect while X et al fed bees with 5 ppb and found no effect; this would be informative if the exposure time and route was the same, but in most cases it is not. I strongly recommend that authors explicitly mention exposure route and duration because they obviously influence the degree / size of the effect. Eg: X et al fed bees 10 ppb (30 DAYS IN POLLEN) of a pesticide and found an effect while X et al fed bees with 5 ppb (60 DAYS IN NECTAR) and found no effect

---

## Round 0.3 · accepted · Accept

Thank you for your response and final revisions on this MS. The MS is now acceptable for publication in PeerJ.

Particular thanks also go to the three reviewers for their work and constructive comments on this MS.